# CRISP: Calibrated Robust Interface-aware Segmentation for Lightweight Polyp Segmentors via Amortized Boundary Posterior Projection under Domain Shift

**Ngoc-Khue Nguyen Vo**[1,2] (iD)                    KHUE.NVN@VINUNI.EDU.VN

**Thanh-Trung Huynh**[3] (iD)                    TRUNG.HT@VINUNI.EDU.VN

**Huy-Hieu Pham**[2,3]                    HIEU.PH@VINUNI.EDU.VN

[1] *Hanoi University of Science and Technology*

[2] *The VinUni-Illinois Smart Health Center*

[3] *VinUniversity*

## Abstract

Polyp segmentation under domain shift fails most critically at lesion boundaries, where geometric errors and unreliable confidence co-occur. We present **CRISP**, a framework for strengthening lightweight polyp segmentors through amortized boundary posterior projection. **CRISP** constructs a boundary-local posterior target from supervision, teacher evidence, and a soft boundary field, then fits it through a calibrated family tied to the student logit. This yields a one-dimensional local projection amortized by a small projector head. Under source-only training on Kvasir-SEG and cross-dataset evaluation on CVC-ColonDB and ETIS, **CRISP** consistently improves boundary geometry and boundary-local calibration on U-Net and PraNet.

**Keywords:** Polyp segmentation, uncertainty calibration, domain shift, boundary-aware.

## 1. Introduction

Polyp segmentation under deployment shift is fundamentally a boundary reliability problem. Clinically consequential errors concentrate where lesion and mucosa meet, where the local decision margin is weakest. Yet the literature still tends to split this problem in two: boundary-aware objectives improve contour fidelity via distance-, contour-, or Hausdorff-style supervision (Caliva et al., 2019; Kervadec et al., 2019; Karimi and Salcudean, 2020), while calibration methods usually act after the decision surface is learned through temperature scaling, uncertainty heads, dropout, or ensembles (Guo et al., 2017; Ding et al., 2021; Wang et al., 2023; Gal and Ghahramani, 2016; Lakshminarayanan et al., 2017; Zhang et al., 2024). Both are useful, but neither directly specifies how calibration should act on the boundary decision surface during learning under shift.

**CRISP** turns calibration into a local posterior projection principle. It constructs a boundary-local posterior target from supervision, teacher evidence, and a soft boundary field, then projects that target onto a calibrated family reachable from the student logit itself. The resulting one-dimensional projection is amortized by a lightweight projector head conditioned on decoder features and logits. In this short paper, we instantiate **CRISP** on lightweight polyp baselines to test whether boundary-aware calibration can materially strengthen efficient segmentors under shift.

## 2. Methodology

Let $x \in \mathbb{R}^{H \times W \times 3}$ be an endoscopic image with binary mask $y \in \{0,1\}^{H \times W}$ and pixel domain $\Omega$. A student network produces a logit map $z(u)$ and decoder features $F(u)$ for each pixel $u \in \Omega$. Figure 1 summarizes the train-time projection path and the student-only inference path of **CRISP**.

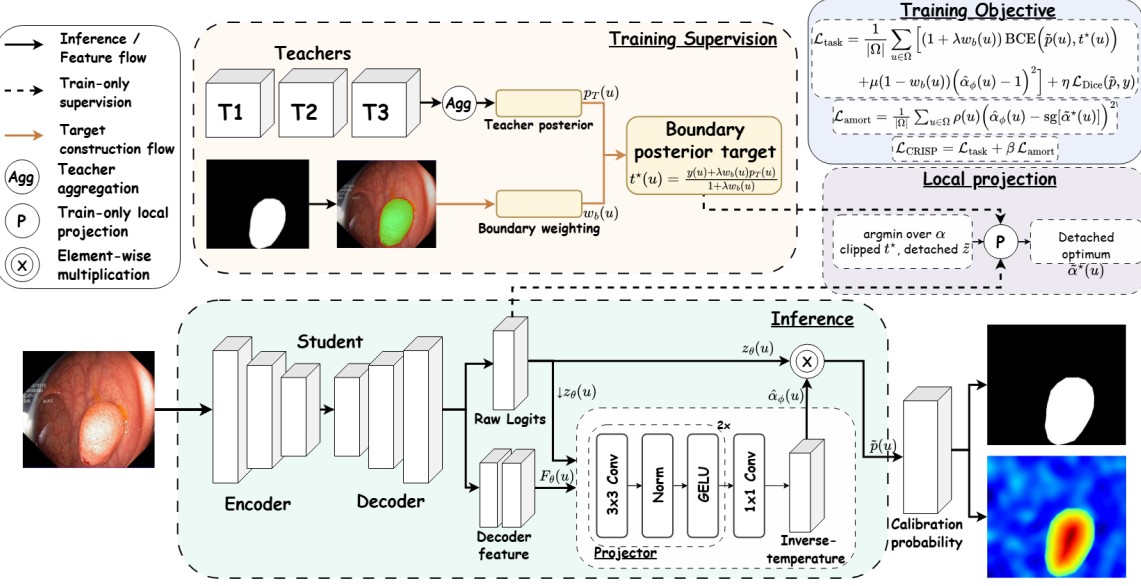

Figure 1: **CRISP pipeline.** Training uses teacher-guided boundary posterior projection, while inference keeps only the student and amortized projector for calibrated segmentation.

**Boundary-local target and local projection.** **CRISP** first localizes where reliability matters most. From the annotation boundary $\partial y$, it forms a soft boundary field $w_b(u) = \exp\left(-\frac{d(u, \partial y)^2}{2\sigma_b^2}\right)$ (1), where $d(\cdot, \partial y)$ is Euclidean distance and $\sigma_b$ controls spread. A teacher pool provides a posterior $p_T(u) \in [0,1]$ during training, and the boundary-local posterior target is $t^\star(u) = \frac{y(u) + \lambda w_b(u) p_T(u)}{1 + \lambda w_b(u)}$ (2), with $\lambda \geq 0$. Hence $t^\star(u) \approx y(u)$ away from the interface, while near it the target becomes a teacher-informed soft posterior localized exactly where boundary ambiguity matters. Rather than learn an arbitrary uncertainty map, **CRISP** restricts calibrated predictions to $\mathcal{Q}(z(u)) = \{\sigma(\alpha z(u)) : \alpha \in [\alpha_{\min}, \alpha_{\max}]\}$ (3), with $0 < \alpha_{\min} < 1 < \alpha_{\max}$, and defines the ideal local inverse-temperature by $\alpha^\star(u) = \arg\min_{\alpha \in [\alpha_{\min}, \alpha_{\max}]} \left[(1 + \lambda w_b(u)) \, \ell(\alpha; z(u), t^\star(u)) + \mu(1 - w_b(u))(\alpha - 1)^2\right]$ (4), where $\ell(\alpha; z, t) = -t \log \sigma(\alpha z) - (1 - t) \log(1 - \sigma(\alpha z))$ and $\mu \geq 0$ preserves identity away from the boundary. This is the scientific core of **CRISP**: calibration is forced to occur along probabilities reachable from the student logit itself, so the method acts on local margins rather than only on post-hoc confidence. The appendix gives the stabilized formulation, the uniqueness result, and the closed-form seed for the detached projection target.

**Amortized projector and training dynamics.** Rather than solving Eq. (4) online, **CRISP** amortizes the projection with a lightweight head $a_\phi$, $\hat{\alpha}_\phi(u) = \alpha_{\min} + (\alpha_{\max} - \alpha_{\min})\, \sigma\big(a_\phi(F(u), z(u))\big)$ (5), yielding the calibrated probability $\tilde{p}(u) = \sigma\big(\hat{\alpha}_\phi(u)z(u)\big)$ (6). Training minimizes $\mathcal{L}_{\text{task}} = \frac{1}{|\Omega|}\sum_u \big[(1 + \lambda w_b(u))\mathrm{BCE}(\tilde{p}(u), t^\star(u)) + \mu(1 - w_b(u))(\hat{\alpha}_\phi(u) - 1)^2\big]$ (7) and $\mathcal{L}_{\text{amort}} = \frac{1}{|\Omega|}\sum_u \rho(u)\, \|\hat{\alpha}_\phi(u) - \mathrm{sg}[\tilde{\alpha}^\star(u)]\|^2$ (8), so $\mathcal{L}_{\text{CRISP}} = \mathcal{L}_{\text{task}} + \beta\mathcal{L}_{\text{amort}}$ (9). For numerical stability, the detached optimum $\tilde{\alpha}^\star(u)$ is computed from a clipped target $t^\star_\varepsilon(u) = \mathrm{clip}(t^\star(u), \varepsilon, 1 - \varepsilon)$ and a detached stabilized logit; this affects only the local solver, not the forward calibrated probability. At the gradient level, $\frac{\partial}{\partial z(u)}\mathrm{BCE}\big(\tilde{p}(u), t^\star(u)\big) = \hat{\alpha}_\phi(u)\big(\tilde{p}(u) - t^\star(u)\big)$ (10), showing that **CRISP** changes both the local target and the strength of boundary updates.

## 3. Experimental Results

Table 1: Cross-dataset robustness on polyp segmentation. Means over five seeds.

| Method | CVC-ColonDB | | | | ETIS | | | |
|---|---|---|---|---|---|---|---|---|
| | Dice | B-F1 | HD95↓ | bECE↓ | Dice | B-F1 | HD95↓ | bECE↓ |
| U-Net | 0.744 | 0.691 | 18.4 | 0.094 | 0.689 | 0.623 | 23.7 | 0.118 |
| U-Net + CRISP | **0.781** (+0.037) | **0.747** (+0.056) | **14.7** (-3.7) | **0.038** (-0.056) | **0.735** (+0.046) | **0.692** (+0.069) | **18.8** (-4.9) | **0.050** (-0.068) |
| PraNet | 0.779 | 0.728 | 15.8 | 0.079 | 0.727 | 0.671 | 20.4 | 0.101 |
| PraNet + CRISP | **0.812** (+0.033) | **0.772** (+0.044) | **12.9** (-2.9) | **0.031** (-0.048) | **0.768** (+0.041) | **0.724** (+0.053) | **16.1** (-4.3) | **0.043** (-0.058) |

We train on Kvasir-SEG (Jha et al., 2020) using source-only tuning and evaluate under cross-dataset shift on ColonDB and ETIS, using two standard backbones, U-Net (Ronneberger et al., 2015) and PraNet (Fan et al., 2020). We report Dice, Boundary-F1 (B-F1), HD95, and boundary-ECE (bECE), which evaluates calibration only on boundary-heavy support. Across both lightweight baselines and both target datasets, gains concentrate most strongly on the boundary-sensitive metrics that **CRISP** is designed to improve, as shown in Table 1. On CVC-ColonDB, **CRISP** raises B-F1 by +0.056 and lowers bECE by 0.056 for U-Net, while improving PraNet by +0.044 B-F1 and lowering bECE by 0.048. The same pattern holds on the harder ETIS shift, where **CRISP** improves U-Net by +0.069 B-F1 and −0.068 bECE, and improves PraNet by +0.053 B-F1 and −0.058 bECE. The gains are therefore concentrated at the lesion interface, where calibration and geometry must remain aligned.

## 4. Discussion and Conclusion

**CRISP** is a compact framework for improving lightweight polyp segmentors under domain shift. Its scientific core is a boundary posterior projection over a calibrated family tied to the student logit, which turns calibration into training-time boundary margin shaping rather than post-hoc confidence repair. Rather than redesigning the segmentor itself, **CRISP** strengthens lightweight baselines such as U-Net and PraNet where deployment failures matter most: the lesion interface. Future work will study richer calibrated families beyond scalar inverse-temperature scaling, allowing the same projection principle to capture more flexible boundary behavior. Another important direction is reducing dependence on external teacher pools through self-ensembling or teacher-free variants, which could broaden the practical reach of boundary-aware calibration under shift.

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

## Appendix A. Theoretical analysis

For the detached local solver, we use the clipped target $t_\varepsilon^\star(u) = \text{clip}(t^\star(u), \varepsilon, 1 - \varepsilon)$ and a detached stabilized logit $\tilde{z}(u)$ satisfying $|\tilde{z}(u)| \in [\zeta, Z_{\max}]$. These stabilized quantities are used only to compute the detached projection target $\tilde{\alpha}^\star(u)$ and do not alter the forward calibrated probability $\tilde{p}(u)$.

**Proposition 1 (Exact boundary-posterior reduction)** *For a fixed pixel $u$, define the local core objective*

$$\mathcal{L}_{\text{core}}^{(u)}(\alpha) = \ell(\alpha; z(u), y(u)) + \lambda w_b(u) \, \text{KL}\big(p_T(u) \,\|\, \sigma(\alpha z(u))\big) + \mu(1 - w_b(u))(\alpha - 1)^2. \quad (11)$$

Then, up to an additive constant independent of $\alpha$,

$$\mathcal{L}_{\text{core}}^{(u)}(\alpha) \equiv (1 + \lambda w_b(u))\ell\big(\alpha; z(u), t^\star(u)\big) + \mu(1 - w_b(u))(\alpha - 1)^2, \tag{12}$$

where $t^\star(u)$ is given by Eq. (2).

**Proof** Let $q(\alpha) = \sigma(\alpha z)$. Since

$$\text{KL}(p\|q) = -p \log q - (1 - p) \log(1 - q) - H(p),$$

substituting $q(\alpha)$ into the local objective yields

$$\mathcal{L}_{\text{core}}^{(u)}(\alpha) = -y \log q - (1 - y) \log(1 - q) \tag{13}$$
$$- \lambda w p_T \log q - \lambda w(1 - p_T) \log(1 - q) \tag{14}$$
$$+ \mu(1 - w)(\alpha - 1)^2 + \text{const}, \tag{15}$$

where $w = w_b(u)$ and $p_T = p_T(u)$. Collecting coefficients of $\log q$ and $\log(1 - q)$ gives Eq. (12) with

$$t^\star(u) = \frac{y + \lambda w p_T}{1 + \lambda w}.$$

∎

**Proposition 2 (Strong convexity and uniqueness of the local projection)** *Assume*

$$0 < \alpha_{\min} < 1 < \alpha_{\max} < \infty, \qquad 0 < \varepsilon < \tfrac{1}{2}, \qquad 0 < \zeta \le Z_{\max} < \infty.$$

*Then the stabilized local projection objective*

$$\tilde{\mathcal{L}}_{\text{proj}}^{(u)}(\alpha) = (1 + \lambda w_b(u))\ell(\alpha; \tilde{z}(u), t_\varepsilon^\star(u)) + \mu(1 - w_b(u))(\alpha - 1)^2 \tag{16}$$

*is strongly convex in $\alpha$ on $[\alpha_{\min}, \alpha_{\max}]$. Consequently, the stabilized projection operator*

$$\tilde{\alpha}^\star(u) = \arg \min_{\alpha \in [\alpha_{\min}, \alpha_{\max}]} \tilde{\mathcal{L}}_{\text{proj}}^{(u)}(\alpha) \tag{17}$$

*is well-defined and unique for every pixel.*

**Proof** Differentiating gives

$$\frac{\partial \tilde{\mathcal{L}}_{\text{proj}}^{(u)}}{\partial \alpha} = (1 + \lambda w)(\sigma(\alpha \tilde{z}) - t)\tilde{z} + 2\mu(1 - w)(\alpha - 1),$$

where $t = t_\varepsilon^\star(u)$ and $w = w_b(u)$. Differentiating again yields

$$\frac{\partial^2 \tilde{\mathcal{L}}_{\text{proj}}^{(u)}}{\partial \alpha^2} = (1 + \lambda w)\sigma(\alpha \tilde{z})(1 - \sigma(\alpha \tilde{z}))\tilde{z}^2 + 2\mu(1 - w).$$

Because $\alpha \in [\alpha_{\min}, \alpha_{\max}]$ and $|\tilde{z}| \in [\zeta, Z_{\max}]$, the scalar $\alpha \tilde{z}$ lies in a compact interval, so

$$c_\sigma := \min_{|a| \le \alpha_{\max} Z_{\max}} \sigma(a)(1 - \sigma(a)) > 0.$$

Hence

$$\frac{\partial^2 \tilde{\mathcal{L}}_{\text{proj}}^{(u)}}{\partial \alpha^2} \geq (1 + \lambda w)c_\sigma \zeta^2 + 2\mu(1 - w) > 0,$$

which implies strong convexity and uniqueness. ∎

**Corollary 3 (Closed-form seed without identity regularization)**  *If $\mu = 0$, the unconstrained minimizer of the stabilized local objective is*

$$\alpha_{\text{unc}}^\star(u) = \frac{\text{logit}(t_\varepsilon^\star(u))}{\tilde{z}(u)}.$$

*The constrained solution on $[\alpha_{\min}, \alpha_{\max}]$ is therefore*

$$\tilde{\alpha}^\star(u) = \Pi_{[\alpha_{\min}, \alpha_{\max}]}\left(\frac{\text{logit}(t_\varepsilon^\star(u))}{\tilde{z}(u)}\right).$$

**Proof**  When $\mu = 0$, the first-order condition becomes $\sigma(\alpha\tilde{z}) = t_\varepsilon^\star$. Applying the logit transform to both sides yields the unconstrained solution, and projection onto the feasible interval gives the constrained one. ∎

## Appendix B. Evaluation metrics

**Dice and boundary geometry.**  We report Dice, Boundary-F1 (B-F1), and HD95. Dice measures region overlap. B-F1 measures contour quality by matching predicted and reference boundary pixels within a tolerance radius. HD95 measures the 95th-percentile Hausdorff distance between predicted and reference boundaries, reducing sensitivity to extreme outliers while still emphasizing geometric boundary errors.

**Boundary-aware calibration.**  For a calibrated probability map $\tilde{p}_i \in [0, 1]$, predicted label $\hat{y}_i = \mathbf{1}\{\tilde{p}_i > 0.5\}$, confidence $c_i = \max(\tilde{p}_i, 1 - \tilde{p}_i)$, and correctness indicator $a_i = \mathbf{1}\{\hat{y}_i = y_i\}$, the standard expected calibration error is

$$\text{ECE} = \sum_{m=1}^{M} \frac{|G_m|}{|\Omega|} \left|\text{acc}(G_m) - \text{conf}(G_m)\right|,$$

where $G_m = \{i \in \Omega \mid c_i \in I_m\}$ are confidence bins. Global ECE can appear overly optimistic in segmentation because easy background pixels dominate the evaluation. We therefore define a boundary support set using the same weighting field that drives the method. For each test image $n$, let

$$\Omega_b^{(n)} = \{i \in \Omega^{(n)} : w_b^{(n)}(i) \geq \tau_{0.8}^{(n)}\},$$

where $\tau_{0.8}^{(n)}$ is the 80th percentile of the boundary weights in that image. Pooling all such pixels gives $\Omega_b = \cup_n \Omega_b^{(n)}$. The boundary-restricted bins are $B_m = \{i \in \Omega_b \mid c_i \in I_m\}$, and boundary-ECE is

$$\text{bECE} = \sum_{m=1}^{M} \frac{|B_m|}{|\Omega_b|} \left|\text{acc}(B_m) - \text{conf}(B_m)\right|.$$

This directly evaluates whether confidence is calibrated where the clinically consequential ambiguity lies.

## Appendix C. Instantiation on lightweight baselines

**Why lightweight baselines.** The short paper focuses on U-Net and PraNet because they are lightweight, familiar, and practically relevant baselines for polyp segmentation. This makes them an appropriate test bed for the question asked by **CRISP**: can boundary-posterior projection systematically strengthen lightweight segmentors under domain shift without redesigning the macro-architecture?

**Practical attachment.** In both cases, **CRISP** attaches only a small projector head to decoder features and logits. The backbone and decoder remain unchanged. Teachers are used only during training, inference uses the student and the amortized projector alone.

## Appendix D. Causal-isolation controls

On PraNet→CVC-ColonDB, distance-relaxed soft labels reach Dice/B-F1/bECE of 0.792/0.744/0.051, while a spatial-$\alpha$ head without posterior projection reaches 0.797/0.754/0.039. Both remain below **CRISP** at 0.812/0.772/0.031. This supports the interpretation that the observed gain is not explained by target softening alone or by adding spatial calibration capacity; the strongest result requires coupling boundary-posterior construction with amortized restricted-family projection.

