# OpenReview forum: "CRISP: Calibrated Robust Interface-aware Segmentation for Lightweight Polyp Segmentors via Amortized Boundary Posterior Projection under Domain Shift"
_MIDL.io/2026/Short_Papers — MIDL 2026 - Short Papers Poster_

### Official Review · Reviewer_cohm · 2026-05-04
**promising but barely understandable paper**

**Rating:** 2
**Confidence:** 4

**Review:**

see strengths and weaknesses below.

**Summary:**

This paper presents a calibration-based method to enhance results of polyp segmentation under domain shift. They use a teacher-student network to gain insight into where and how uncertainty is located and to calibrate the model's predictions with a projection head.
The method is then tested on two out-of-domain datasets, where it outperforms Unet and PraNet simple baselines in terms of segmentation accuracy.

**Strengths:**

1. There seems to be novelty in the proposed framework, in particular with the projection head and the way calibration is performed.
2. The method is tested on 2 out-of-domain datasets, where it produces good results (but against weak baselines).
3. This paper tackles an interesting problem, where calibration is key for clinical deployment and decision-making.

**Weaknesses:**

1. The method is pretty convoluted and making it fit on a 3-page paper was done by sacrificing a lot of clarity: there are no explanations to motivate the many subparts of the methods, many notations are not introduced, multi-line equations are hard to follow, the figure is extremely unclear, etc. Moreover, the paper is not self-contained and many details are out-sourced to long appendices. As a result, this paper is barely understandable, even after spending considerable time re-reading the paper.

2. The experimental framework is a bit light. First, the authors only test their method against 2 very standard/weak baselines (Unet, PraNet), which do not include any calibration mechanism. Therefore, it is impossible to position this work against state-of-the-art calibration-aware approaches. Moreover, ablation studies are imperative for this type of framework to judge the relevance of the many proposed modules.

3. Relatedly, the experiments are poorly explained. What exactly are these datasets, and what does the domain shift consists in: intensity distribution? anatomy? acquisition parameters? Also, no details are given about implementation of the methods and the baselines: what architecture? any augmentation? data pre-processing? hyper-parameters (notably the alphas)?

4. Albeit being central to the paper, calibration is poorly introduced, and not motivated at all.

**Justification Of Rating:**

While the paper seems to be interesting, the current form doesn't allow for a any comprehension under the 3-page format. Moreover the experimental setup is very weak and doesn't allow for positioning against the calibration literature and does not disambiguate the contribution of each sub-module.

---

### Decision · Program_Chairs · 2026-05-08

Accept (Poster)